# Cutting Edge Aquatic-Based Collagens in Tissue Engineering

**DOI:** 10.3390/md21020087

**Published:** 2023-01-26

**Authors:** Jonathan Ardhianto Panggabean, Sya’ban Putra Adiguna, Mutia Hardhiyuna, Siti Irma Rahmawati, Nina Hermayani Sadi, Gunawan Pratama Yoga, Eva Nafisyah, Asep Bayu, Masteria Yunovilsa Putra

**Affiliations:** 1PT. Biotek Rekayasa Indonesia, Jalan Perserikatan No. 1, Rawamangun, Pulo Gadung, Jakarta Timur, Daerah Khusus Ibukota, Jakarta 13220, Indonesia; 2Research Center for Vaccine and Drugs, National Research and Innovation Agency, Jalan Raya Jakarta Bogor KM. 46, Cibinong, Bogor 16911, Indonesia; 3Research Center for Limnology and Water Resources, National Research and Innovation Agency, Jalan Raya Jakarta Bogor KM. 46, Cibinong, Bogor 16911, Indonesia

**Keywords:** collagen, aquatic-based collagen, collagen scaffolds, tissue engineering, tissue regeneration

## Abstract

Aquatic-based collagens have attracted much interest due to their great potential application for biomedical sectors, including the tissue engineering sector, as a major component of the extracellular matrix in humans. Their physical and biochemical characteristics offer advantages over mammalian-based collagen; for example, they have excellent biocompatibility and biodegradability, are easy to extract, and pose a relatively low immunological risk to mammalian products. The utilization of aquatic-based collagen also has fewer religious restrictions and lower production costs. Aquatic-based collagen also creates high-added value and good environmental sustainability by aquatic waste utilization. Thus, this study aims to overview aquatic collagen’s characteristics, extraction, and fabrication. It also highlights its potential application for tissue engineering and the regeneration of bone, cartilage, dental, skin, and vascular tissue. Moreover, this review highlights the recent research in aquatic collagen, future prospects, and challenges for it as an alternative biomaterial for tissue engineering and regenerative medicines.

## 1. Introduction

Tissue or organ transplantation has been accepted as a therapy option for patients with organ or tissue failure [1]. More than 34,285 solid organ transplantations in Europe were performed in 2019 [2]. Physicians transplant organs from one individual into another by performing surgical reconstruction or using mechanical devices. Although these therapies are safe, they are limited by a lack of potential donors [3,4]. Moreover, the possible microbial exposure, infections, and tissue damage on the outcome of solid-organ transplantation should not be ignored due to the release of pathogen- and damage-associated molecular patterns, as well as pathogen- or allograft-derived antigens [5]. The microbiota, i.e., the communities of microorganisms that colonize mucosa and epithelial surfaces, could also modulate alloimmunity, accelerate transplant rejection, and affect immunosuppressive therapy [6].

Tissue engineering has attracted much attention as a proper method in the field of pathologies for treating tissue damage. This interdisciplinary field applies the principles of engineering, biochemistry, and life sciences to the development of biological substitutes that can mimic, restore, maintain, or improve original tissues and their function [3,7]. The basic principle of tissue engineering is using biodegradable and biocompatible polymers, viz., exogeneous three-dimensional extracellular matrices (ECMs), to support the growth of natural new cells [8]. In the 1990s, Langer and colleagues introduced three general concepts for tissue engineering in drug delivery applications involving cells isolated from a small biopsy, tissue-inducing substances, and cells placed on or within matrices [3]. The involved matrices derived from biomaterials are called scaffolds and are further incorporated into the body [9,10]. These scaffolds can mimic the extracellular matrix present in vivo as a supporting structure for cell cultures, temporarily aid cell regeneration, and gradually biodegrade, resulting in natural new tissues that replace lost or malfunctioning tissues or organs (Figure 1) [7,11]. The biodegradability of scaffolds is necessary to eliminate the side effects of foreign materials left in the body [11]. As such, the polymer scaffold is a critical element and should possess specific chemical, mechanical, and physical properties to achieve cell diffusion and three-dimensional (3D) tissue formation, e.g., cell growth, vascularization, proliferation, and host integration, as well as to be naturally degradable during or after the healing process [11]. 

Hydrogels are one of the most prominent and versatile biopolymers for tissue engineering. These materials possess structural similarities to the macromolecular-based components in the body. For instance, they contain high water content (>30% *w*/*w*) and good mechanical flexibility, similar to natural tissues, which support the convenience of molding into any shape or structure [7,12]. The high water-content maintains a biocompatible aqueous condition inside hydrogels for encapsulated cell proliferation, migration, differentiation, and communication, which also protects them from immune attacks, cytotoxic molecules, and mechanical stresses [13]. These properties lead hydrogels to meet the criteria for tissue engineering, including physical properties, e.g., degradability, mechanics, and gel formation; mass transport properties, e.g., diffusion; and biological properties, e.g., cell adhesion and signaling [1,7,12]. 

Collagen is an attractive natural polymer for tissue engineering applications. It is the main component of ECMs on mammalian tissues, including skin, bone, cartilage, tendon, and ligament tissues [1]. The inherent physical and biochemical properties of collagen are suitable for inducing tissue formation [14,15], such as excellent absorbability in the body, low antigenicity, non-toxicity, high tensile strength, high affinity with water, good biocompatibility, and biodegradability [16]. Its low immunogenicity diminishes the possibility that it will be rejected when ingested or injected into a foreign body [17]. Collagen may be used as a sponge-shaped scaffold with a porous 3D structure [18,19] for biomimetic tissue engineering applications, such as bone grafting, cartilage repair or regeneration, osteochondral tissue engineering, and membrane scaffolding for cell attachment. Several physical properties of collagen scaffolds, including their mechanical strength, pore structure, porosity, interconnectivity, and bioactivity, are critical in supporting its function in tissue engineering [20], which is vital for collagen to withstand pressure from the body, as well as for the migration and proliferation of cells and the transfer of nutrients. Recently, nano- and micron-sized fibrous collagen has been widely used in fiber-shaped tissue engineering in, for example, neural pathways, blood vessels, and muscle tissues [21,22].

In 2021, collagen’s global market reached USD 3.6 billion and was predicted to increase by 8.3% until 2027 due to its high demand in the biomedical, cosmeceutical, food, and pharmaceutical industries [23,24]. Generally, the market is dominated by bovine- and porcine-derived collagens (BPCols), which account for almost 35% of the total market [25]. Despite their broad application and long usage history, BPCols have challenges related to their complex tunable mechanical characteristics, sensitivity to enzyme degradation, and low thermal stability [26,27,28,29]. Furthermore, BPCols face the risk of the transmission of bovine spongiform encephalopathy (BSE) [26,27] and foot-and-mouth diseases through indirect contact with an animal product via ingestion or secondary aerosols [28]. Therefore, explorations of sources of collagens other than BPCols have become increasingly common.

Recently, aquatic-based collagens (AQCols) have gained interest in various applications, from food packaging with antibacterial activity [30] to tissue engineering [14]. AQCols are attractive because they provide an opportunity to add value to seafood waste [31,32]. They could also connect the circular economy by maximizing the value of seafood raw materials [31]. In contrast with BPCols, aquatic sources of collagen could minimize the zoonosis risk. Moreover, the use of BPCols is limited in the Muslim community owing to its halal policy. Herein, the utilization of aquatic organisms guarantees the halal quality of the sources, thus providing widely accepted collagen without religious concerns [33,34]. Furthermore, AQCols show comparable characteristics to BPCols. For example, bovine-based and tilapia-skin-based collagen exhibit similar results in the promotion of epidermal growth factor, fibroblast growth factor, and vascular endothelial marker expression [35]. A semi-quantitative in vivo study on AQCol from jellyfish (*Rhizostoma pulmo*) showed a lower histopathology score, including inflammation, fibrosis, necrosis, and neovascularization scores, compared to BPCol. The positive findings were also supported by the absence of a severe immunogenic response from the studied Wistar rats [36]. The promising potential of AQCol to be a versatile and sustainable source of biomaterial must be guided, and the established utilization of BPCol with advanced modification and biocompatibility studies is required.

This review describes the current status of research on AQCols. The structure and function of AQCols are presented, along with a comprehensive discussion of their extraction processes and characteristics. This paper also includes an exploration of various applications of this collagen, particularly regarding its potential application in the regeneration of different tissues. It is expected that this review could give insights into the potential exploration of AQCol in tissue engineering with novel characteristics.

## 2. Aquatic-Based Collagen

The term “collagen” refers to a group of proteins that comprises 30% of the whole protein in animals. The group members consist of a polypeptide sequence with repeating glycine-X-Y units, in which X and Y are commonly found as proline (Pro) and hydroxyproline (Hyp), respectively (Figure 2). The members also share a right-handed triple-helix structure consisting of three α chains resulting from the intertwining polypeptide chains that are held together by a hydrogen bond [37,38]. The three polypeptide fibrils can have a diameter of 10–500 nm, with an approximate molecular weight of 285 kDa and 1400 amino acids [17]. The collagen chain domain is also “knotted” by telopeptides consisting of N-terminus and C-terminus amino acids, with the triple-helix serving as the fibril and establishing the secondary structure of collagen members [39,40]. The C-terminus telopeptides are also involved in the initiation of triple-helix formation, and the N-terminus helps to regulate fibril diameter. In addition, the telopeptides provide cross-linking and linking sites for another molecule on the surrounding matrix (Figure 3) [41,42,43].

Collagen includes 28 protein types, viz., Type I to Type XXVIII, depending on the different distribution ratios in tissue, amino acid composition, and its organization on a supramolecular scale [23,25]. Among them, collagen Type I is the most abundant (60% to 80%), followed by Type II (15% to 20%) [17]. Types I and III are commonly found in skin, tendon, bone, ligament, and interstitial tissues, whereas types II and XI are found in cartilage, intervertebral discs, and bone enamel tissues [45,46,47]. This difference is related to their chemical composition, which also dictates their functionality. For instance, collagen type II contains more glycosylated matter than type I, which influences the variation of their intermolecular and interfibrillar interactions [48].

Despite their similar collagenous type, AQCols and BPCols have different molecular characteristics, particularly in their amino acid composition, which result in different physical characteristics. A recent study suggested that the amino acid content of Pro and Hyp extracted collagen from fish with a warmer habitat (6 species—*Pterocaesio digramma*, *Coryphaena hippurus*, *Pseudorhombus pentophthalmus*, *Pleuronichthys cornutus*, *Scomber australasicus*, and *Decapterus tabl*) is more durable compared to collagen produced from a colder habitat (5 species—*Pleurogrammus azonus*, *Synaphobranchus bathybius*, *Coryphaenoides pectoralis*, *Coryphaenoides acrolepis*, and *Lycenchelys squamosal*). The study also examined the presence of another amino acid, serine (Ser), which increases the flexibility of its fibrous collagen [49]. Another study mentioned the presence of cysteine in collagen extracted from Japanese sea bass, Nile tilapia, and commercial porcine skin. The study revealed the superiority of the mechanical strength and thermal stability of bass and tilapia collagen due to the intermolecular disulfide bonds, in addition to the hydrogen bonding on the intertwining triple-helix structure [50]. Another difference can be seen in the amino acid content of sea cucumber collagen, which contains more Gly and Alanine (Ala) and less Hyp than mammalian collagen. The absence of Hyp causes low hydrogen bonds between hydroxy groups, lowering the denaturation temperature of sea cucumber collagen compared to mammalian collagen [51,52,53].

The development of aquatic collagen presents both opportunities and challenges due to the different characteristics of each source [54]. It affects the properties of AQCols, such as their temperature behavior, which could differ from that of mammalian collagens and even the temperature of the human body. Modifications and fabrications are typically made to improve their characteristics, which is also true of mammalian collagen (Table 1) [55].

### 2.1. Sea Cucumber Collagen

Sea cucumbers could be a potential alternative source of collagen [56] since their body walls are composed of mutable connective tissue consisting of collagen, proteoglycan, and glycoprotein (Figure 4) [53]. Collagens derived from this marine invertebrate have been reported to possess high moisture retention and absorption capacities; they also exhibit bioactive properties, e.g., radical scavenging and angiotensin I converting enzyme (ACE) inhibition [57,58,59]. Sea cucumber’s collagens are a type I and consist of three α1 chains with a triple helical structure, which may be found as one band in sodium dodecyl-sulfate polyacrylamide gel electrophoresis characterization (SDS-PAGE) [51,60,61]. Their amino acid compositions mainly comprise Gly, Pro, Ala, and Hyp, with variations of glutamic acid (Glu), threonine (Thr), and Ser, depending on the species [62,63,64]. These amino acids could distinguish sea cucumber collagen from mammalian collagen [65]. Herein, collagens derived from sea cucumber contain low Hyp, with high Glu and aspartic acid (Asx) residues, which contrasts with mammalian collagens.

The amino acid composition highly influences the physical and chemical properties of collagen, such as its thermal stability [51]. The low Hyp content of sea cucumber collagen makes its thermal stability relatively low because it forms fewer hydrogen bonds from its hydroxyl groups than mammalian collagen [51,59,60,67,68]. For example, collagen isolated from *Stichopus monotuberculatus* had a maximum transition temperature (T_m_) of 30 °C, while calfskin collagen had a T_m_ of 35–41 °C [53]. Collagen derived from *Stichopus japonicus* and pepsin-solubilized collagen (PSC) from *Holothuria parva* were reported to be 57 °C and 47 °C, respectively, which were lower than type I collagen from bovine skin (62 °C) [63]. The denaturation temperature (T_d_) values show similar trends: they were around 18 °C, which is lower than that obtained from porcine skin (37 °C) [60]. Therefore, PSCs generally have poorer thermal behavior than their original collagen fibrils. In addition, collagens of sea cucumber are considered bipolar collagen fibrils with surface-associated proteoglycans [53].

**Table 1 marinedrugs-21-00087-t001:** Terrestrial-inspired modification of AQCol for biomimetic tissue engineering.

Collagen Source	Application	Fabrication Method	Cross-Linker	Remarks	Ref.
Atlantic salmon skin	Bone tissue engineering	Controlled freeze-drying and chemical cross-linking	EDC and HCl	(1) Stable under cyclic compression (2) Comparable elastic mechanical properties to bovine collagen	[69]
Mineralized salmon collagen	Tissue engineering	Freeze-drying	n/a	(1) Addition of aloe vera decreased the mean pore size, increased porosity, and changed the pore architecture. (2) Reduced tensile strength (3) Increased temperature of dehydration (Td)	[70]
Fish scale (*Larimichthys crocea*)	Topical wound healing	Hydrogel cross-linking through Schiff base reaction	Oxidized sodium alginate	(1) Increasing tensile strength to the same level as human skin (2) Increased the G’ (3) Enduring external compression	[71]
Fish skin (*Oreochromis* *niloticus*)	Cartilage repair	Freeze-drying technique	Carbodiimide	(1) Increased density (2) Reduced pore size and porosity	[72]
Blue shark skin (*Prionace glauca*)	Cartilage tissue regeneration	Cryogelation with Hya	EDC and HCl	(1) High stability of cryogels G’ (Coll > Coll: Hya) (2) Water uptake (Coll < Coll:Hya) (3) Pores and interconnectivity (Coll < Coll:Hya)	[73]
	Hard tissue engineering	In situ mineralization	Calcium-phosphorous	Viscosity and G’ of mineralized collagen (alginate < alginate bioink)	[74]
Jellyfish and salmon collagen	Osteochondral tissue engineering	Freeze-dryingand crosslinking	n/a	Stable under cell culture conditions without any delamination	[18]
Salmon collagen	Thermal stability improvement	UV irradiation	n/a	Increased thermal stability	[75]
		Fibril formation and cross-linking	EDC and HCl	Increased thermal stability from 18.6–47 °C	[76]
Blubber jellyfish (*Catostylus mosaicus*)	MC3T3 preosteoblast cell attachment and proliferation promotion	Fibril formation using Tris-buffer	n/a	(1) Increased thermal transitions at ~55 °C (2) Slight increase in G’ (collagen agarose) (3) Collagen agarose is 50-fold stiffer than pure Jellagen	[77]
Barrel jellyfish (*R. pulmo*)	Matrix for chondrocyte embodiment	Hydrogelation	Genipin	(1) Denaturation temperature increased to 57 °C (2) Rheological properties show a crossing point between G’ and G” at ~100% strain. (3) Constituted a suitable matrix for human chondrocyte embodiment	[78]
	Cartilage repair	Freeze-drying	EDC and HCl	No mechanical improvements	[79]
Flame jellyfish (*Rhopilema* *esculentum*)	Matrix for chondrocyte transplantation	Freeze-dried using polyoxymethylene casting molds	EDC and HCl	(1) Lengthy pores with a honeycomb structure (2) Increased thermal stability to above 50 °C (3) Increased stiffness comparable with human placenta collagen	[80]
East Atlantic jellyfish (*Catostylus tagi*)	Protein delivery	Emulsification-gelation-solvent extraction	EDC and HCl	CMPs—moderate hydrophobic behavior and a positive surface charge. Increased stability in water, allowing a slow release	[81]
Sea cucumber (*Stichopus hermanii*)	Burn wound treatment	Hydrogelation at RT with gamma irradiation	n/a	Significant increase in the healing rate in a rat-burning wound.	[82]
Sea cucumber (*Holothuria* *tubulosa*)	Guided tissue regeneration	Chemical cross-linking	EDC/NHS	(1) Higher stiffness and tensile strength (20-fold) than mammalian collagen (2) Thinner membrane (3) Higher mechanical resistance than commercial membranes	[66]
Sea cucumber (*Apostichopus* *japonicus*)	Food stability improvement	Chemical cross-linking	AG and CA	(1) Increased hardness by 108% (AG) and 254% (CA) at 30 days (2) Smaller pore size with finer collagen fibrils (3) Inhibited the breakage of peptide bonds in RSC collagen	[83]
Sponge (*Ircinia fusca*)	Bone grafting	Freeze-drying and lyophilization method biomineralization	Chitosan hydroxyapatite	(1) High thermal stability composite (2) Interconnected porosity (3) In vitro cell proliferation	[84]
Sponge (*Chondrosia* *reniformis*)	Tissue engineering and regenerative medicine	Chemical cross-linking	EDC/NHS	(1) Good mechanical properties, enzymatic degradation resistance, water binding capacity, antioxidant activity, and biocompatibility on both fibroblast and keratinocyte cell cultures (2) Low viscosity and typical gel behavior	[85]
Sponge (*Aplysina fulva*)	Bone tissue engineering	Silicon mold with pressurized air	Hydroxyapatite	Improved biological properties for mimicking bone graft regeneration	[86]
Sponge (*C. reniformis*)	Collagen nanoparticles andtransdermal delivery	Nanoparticle by controlled-alkaline hydrolysis	Polymethacrylate	Spherical nanoparticles enabled a prolonged estradiol release compared to a commercial gel	[87]
		Estradiol-hemihydrate loaded by adsorption			
		Emulsifying and cross-linking	Glutaraldehyde	Increased penetration of trans-retinol against the skin (approximately 2-fold) compared to a gel without collagen nanoparticles	[88]
Sea urchin (*Paracentrotus lividus*)	Tissue regeneration	Chemical cross-linking	EDC/NHS	Provided cells with a biomimetic environment in terms of structure, biochemical composition, and mechanical characteristics	[89]

EDC—1-ethyl-3-(3-dimethylaminopropyl)-carbodiimide; NHS—N-hydroxysuccinimide; Hya—hyaluronic acid; G’—storage modulus; G”—loss modulus; CMPs—collagen microparticles; AG—*Apium graveolens* (celery); CA—chlorogenic acid.

In addition to the amino acid composition, sea cucumber collagen may be an alternative collagen source through SDS-PAGE. This potential of sea cucumber collagen relies on the different bands that emerge from SDS-PAGE analysis, which show a distinct 3 α1-chain between 75 and 100 kDa for the collagen extracted from *H. cinerascens*’s body wall compared to the similar tilapia and commercial porcine skin collagen (Figure 5A) [90]. The distinction also comes from other research that shows a thick band of an α-chain with various molecular weights from slightly above 75 kDa [90], slightly above 116 kDa [60], and 116–200 kDa [51] (Figure 5). Unfortunately, it is found that the unique collagen architecture of sea cucumber shows poor thermal stability compared to porcine and tilapia skin collagen [90]. Despite the poor thermal stability, the distinction of the protein architecture might be useful in specific applications, especially with tunable modification. 

The fabrication of biomaterials might be essential to achieving a suitable sea cucumber collagen for biological purposes, which is urgently needed, considering that its physical and chemical properties are not sufficient to accomplish the mammalian collagen standard for application in human beings. Moreover, sea cucumber has an endogenous enzyme (endogenous cysteine proteinases and serine proteinase) that is activated as the temperature rises (15–30 °C), thereby gradually decreasing the shear-thinning and pseudo-plastic properties, as well as the visco-elasticity, due to the degradation of interfibrillar proteoglycan bridges in the collagen fibers [91,92,93]. Biomimetic fabrications, such as gels, fibers, and membranes, and the cross-linking of sea cucumber-based collagen have more resistant mechanical properties than mammalian-based collagen. Membrane fabrication using sea cucumber-based collagen can improve its mechanical properties, including its stiffness and tensile strength, up to 20-fold compared to mammalian collagen [66]. Some cross-linking techniques—such as the use of the gamma irradiation technique (physically) and EDC/NHS, *Apium graveolens* (celery) extract, and chlorogenic acid (chemically)—have been studied and appear to yield increases in hardness of 108% or 254% compared to uncross-linked collagen. In addition, cross-linked collagen exhibits finer and smaller collagen fibrils and inhibits the breakage of peptide bonds in sea cucumber collagen [82,83].

Future applications of sea cucumber-based collagen as biomaterials and for biomedical uses have remained limited since the raw material for sea cucumbers is expensive. However, this should not be an obstacle in the future because the development of sea cucumber cultivation is already available.

### 2.2. Sea Urchin Collagen

Sea urchins have a circular soft tissue area surrounding a peristomial membrane (PM) that primarily comprises mammalian-like collagen (Figure 5). The PM is a thick layer of skin reinforced by ossicles lined by the epidermis on the outside and coelomic epithelium tissue on the inside. This collagen is similar to mammalian type-I collagen in terms of chain composition, ultrastructure, and immunoreactivity [94,95]. The PMs of edible sea urchins represent a potentially untapped source of collagen that is readily available as a waste product from the food industry. Furthermore, sea urchin collagen is non-toxic because the cells adhered and survived; moreover, they exhibit active proliferation in the medium–to–long term (7–21 days), [89].

Beneditto et al. [89] demonstrated that native collagen fibers extracted from PMs are suitable for in vitro systems. In vitro studies with stem cells show good biocompatibility regarding overall cell proliferation. The results showed that sea urchin *P. lividus* could be a valuable, low-cost source of collagen, especially for tissue engineering resources. Additionally, sea urchins are harvested in several countries (Japan, the United States, and France), potentially enabling an industrial and stable supply of coarse collagen material that biotech companies can use.

### 2.3. Fish Collagen

Fish collagen is a typical source of aquatic collagen [55]. The utilization of fish has many advantages over mammalian tissues in producing collagen. For example, the skin and scales of fish are the only sources of waste in the fish industry, but these low-value materials contain a significant amount of collagen [96]. The waste is a cheap source of high-value collagen. As such, the utilization of by-products of fish could form sustainable fisheries for making multiple valuable products.

The characteristics of fish collagen vary depending on certain factors [14]. For example, salmon collagen has a T_d_ of 18.6 °C [76]. In general, the T_d_ values of fish collagens are around 18–30 °C [15]. Fish collagen is less cross-linked and has lower mechanical strength than bovine collagen. Several modifying processes by chemical cross-linking and other physical fabrication methods have been successfully reported to improve the thermal stability of fish collagen [14]. Hoyer et al. carried out biomimetics to produce mineralized salmon collagen for repairing bone tissue by controlled freeze-drying and chemical cross-linking using EDC [69]. This method obtained a stable collagen scaffold biomaterial under cyclic compression and showed similar elastic mechanical properties as bovine collagen. Fibril formation in the EDC cross-linking process increases salmon collagen’s strength and thermal stability (47 °C) [76]. 

Biphasic scaffolds from biomimetically mineralized salmon collagen and fibrillated jellyfish collagen formed by joint freeze-drying and cross-linking have been reported to increase stability under cell culture conditions without delamination [18]. Moreover, UV irradiation can stabilize the low denaturation temperature of collagen matrices fabricated from salmon atelocollagen [75]. Cryogelation and mineralization applied to a blue shark (*P. glauca*) skin collagen improved the stability of the cryogels and increased the storage modulus, water uptake, viscosity, and pore size [73,74]. This mineralized collagen was the first to be used as a 3D-printable, cell-laden hydrogel from a marine source with a high survival rate of mouse fibroblast cell lines [74].

### 2.4. Jellyfish Collagen

Jellyfish is a scyphozoan class that has spread across the world [97]. Acid-soluble collagen (ASC) from jellyfish was reported to be fibrillar type I collagen with two α1 and one α2 chain and a β-dimer (Figure 6) [36,77]. It shows a similar structure to human type II collagen and provides better support for the chondrogenic phenotype than mammalian collagen [80,98]. The thermal stability of jellyfish collagen is species-dependent and has a melting point of 24–32 °C [77,99,100]. The stabilization of the collagen triple-helix structure is highly affected by the content of Pro and the imino acid position [101].

The collagen extracted from the *Scytphomedusa rhizostoma pulmo* was implanted into a mouse model and showed optimal adsorption and biocompatibility properties [50]. Collagen scaffolds derived from jellyfish mesogloea showed no cytotoxicity and had higher cell viability than other biomaterials (bovine collagen, hyaluronic acid, gelatine, and glucan). Further, the biostability of jellyfish collagen scaffolds and the increased resistance against enzymatic degradation are potentially safe tissue engineering materials [102]. Furthermore, a biocompatibility test was conducted using primary stem cell cultures and mesenchymal stem cells. These findings suggest that collagens derived from jellyfish may become a suitable replacement for bovine-derived collagen due to their excellent biocompatibility.

The natural triple-helix structure of jellyfish collagen is insufficient for biological purposes due to a lack of thermal and rheological properties. Some biomaterial fabrication of jellyfish collagen has been achieved. Incorporating the collagen into agarose exhibited a slightly increased storage modulus and rigidity of the scaffold (almost 50-fold stiffer) compared to pure jellyfish collagen hydrogel and agarose solely [77]. Meanwhile, 3D hydrogelation using genipin, i.e., geniposide, a product isolated from *Gardenia jasminoides* fruit [103], as a natural cross-linker significantly fixed the properties, e.g., it increased the thermal stability of the scaffold to reach 57 °C, thereby enhancing the rheological properties to be more elastic at low strain (<100%) and more viscous at high tension (>100%), compared to pure fibrillar jellyfish collagen [78]. This cross-linked hydrogel is a suitable matrix for human chondrocyte embodiment [78]. Sewig et al. showed a stiffness comparison of jellyfish (*R. esculentum*) collagen with human placenta (type I) collagen according to the EDC concentration as a chemical cross-linking agent. Jellyfish collagen and human placenta collagen showed comparable lengthy pores in a honeycombed structure, similar stiffness (from 10.4 ± 0.6 to 13.4 ± 2.2 kPa), and an increased melting point of more than 50 °C [80]. These results indicate that jellyfish collagen is suitable for biomaterial applications, as well as human-based collagen.

### 2.5. Marine Sponge Collagen

Marine sponges present a variety of secondary metabolites that have broad biological activities, especially as antibacterial and anticancer agents [104]. In addition to providing a variety of secondary metabolites, marine sponges are a potential source of collagen [105]. For example, *C. reniformis* contains collagen up to 30% *w*/*w* [19]. Like other AQCols, sponge collagens have a low thermal stability of around 20 °C. They are rich in acidic amino acids (Asx and Glu) and have a Hyp/Pro ratio of 0.4–0.9 [106,107,108]. These values are similar to marine eel fish, which have been reported to have a Hyp/Pro ratio of 0.98 [109].

Collagens derived from sponges have been used for drug delivery and tissue engineering (bone grafting). Nevertheless, modifications or biomimetics should be made to make these (similar to calf or bovine collagens) [110]. Furthermore, collagen sponges have low toxicity and immunogenicity and are free from BSE transmission [111]. Sponge collagens were successfully formed as nanoparticles for transdermal delivery of estradiol and retinol. The formation process involves glutaraldehyde and polymethacrylate as cross-linkers. Drug delivery can prolong estradiol release and increase trans-retinol penetration into the skin compared to gels without collagen nanoparticles [87,88]. In addition to being a drug delivery agent, marine sponge collagen adapts the application of terrestrial collagen as bone grafting by involving chitosan and hydroxyapatite. The composite formed has high thermal stability with a weight loss temperature of >300 °C (lower than the composite of rat tail tendon, which has a weight loss temperature of 500 °C) [112]. It also has lower porosity and water retention abilities than pure chitosan, which also makes it suitable for bone tissue engineering [84,86,113].

## 3. Extraction and Isolation of AQCol

The process of extracting AQCols is almost the same for all raw materials. The extraction is carried out by removing non-collagenous proteins, pigments, and fats. Generally, the processes can be divided into two main groups: conventional and assisted extraction (Figure 7). Conventionally, acid/alkali and salt are used as hydrolyzing and solubilizing agents. Meanwhile, assisted extraction involves external energy or enzymes to optimize the extraction yield. Primarily, the extractions are conducted under a controlled temperature of 4 °C to protect the collagen against thermal degradation [55,114,115]. Nevertheless, the efficiency of the process is affected by several factors, including the raw material pre-treatment, storage condition, and extraction processes. In addition, the functional properties of the isolated collagen, e.g., the polypeptide chain length, viscosity, and solubility, are affected [116].

Raw materials contain non-collagenous proteins, pigments, lipids, fats, and inorganic materials, such as calcium and other minerals. Chemicals such as ethylenediaminetetraacetic acid (EDTA) or HCl are used to remove inorganic materials. Meanwhile, non-collagenous proteins, pigments, and fats are removed by a basic or saline solution (NaOH/NaCl), hydrogen peroxide, and alcohol (n-butanol), respectively [117,118,119,120]. The application of these treatments depends on the composition of the raw material. Fish bones and scales, for example, contain many minerals, and the material must be demineralized using EDTA. Meanwhile, skin and ligaments holding non-collagenous proteins, fats, and alkaline or saline solutions should be applied [121,122,123,124,125].

### 3.1. Conventional Method

Collagens are typically extracted using acid solutions [115]. This process breaks the intermolecular bonds between collagen molecules to obtain ASC fibrils. Acidic solutions promote a positive charge and increase the repulsion between tropocollagen molecules, increasing collagen’s solubility. Most acid hydrolysis methods use 0.5 M acetic acid, and the reaction is stirred continuously for 24–72 h. Later, NaCl is added to precipitate the crude collagen fibrils. The high-quality collagen could be purified from the ionic solution using dialysis in 0.1 M acetic acid and distilled water for 2 days under a controlled temperature.

The selection of acid solution affects the quantity and properties of collagens. Skierka and Sadowska reported that about 90% collagen yield was obtained from Baltic cod (*Gadus morhua*) skin using acetic and lactic acid solutions, while citric and hydrochloride acids gave 60% and 18% yield, respectively [126]. They also mentioned that additional enzymatic (pepsin) treatment and these acids could significantly improve collagen yield. The acidic treatment under high temperatures leads to a low molar mass and firmer gel formation, implying the significance of setting the extraction temperature at 4–20 °C to prevent collagen decomposition [127]. Naturally, collagen causes autolysis due to the endogenous enzyme. Therefore, the extraction temperature is an essential factor that directly affects collagen yield. Some reports have also described the effectiveness of alkaline solutions, including calcium oxide, calcium hydroxide, and sodium carbonate, as collagen extraction media [128]. However, this condition could be a potent hydrolyzing agent, possibly destroying some amino acids, such as serine, cysteine, histidine, and threonine [129].

Alternatively, collagen extraction has been reported to be carried out using salt (NaCl) to obtain salt-soluble collagen (SSC). However, this is ineffective because collagen molecules are less soluble in salt and neutral solutions. The solubility of type I collagen is observed to be less than 1.0 kmol·m^−3^ and precipitates at concentrations exceeding 1.0 kmol·m^−3^ at specific salt concentrations [53,115]. For example, SSC from sturgeon (*Acipenser schrenckii*) cartilage and skin have been isolated using 0.45 M NaCl, yielding 2.18% and 4.55%, respectively. These results are insignificant compared to ASC and PSC from the same source, as ASC of cartilage and skin was 27.04 and 37.42%, while PSC was 55.92 and 52.80%, respectively. It should be noted that SSC’s denaturation and melting temperatures are lowest compared to ASC and PSC [130,131].

### 3.2. Assisted Extraction

In the assisted extraction process, the addition of external energy, such as ultrasound, microwave, and enzymes-assisted energy, is combined with conventional methods to propose a hybrid method to maximize the extraction process and increase the amount and purity of collagen obtained. Therefore, the extraction process is suitable for industrial applications [115]. Nevertheless, the use of ultrasound and microwave energy typically results in high heat at a molecular level if these processes are conducted over a long time. As such, the operation condition should be strictly controlled to prevent the denaturation of the extracted collagen. In this section, such procedures are discussed in detail.

#### 3.2.1. Enzyme-Assisted Extraction

AQCol can be obtained by applying acid-enzyme extraction [51,52,56,60,92,132]. Enzymatic-assisted extraction could benefit from increasing the specificity and yield of extraction by shortening the extraction time [53]. Enzymatic hydrolysis is highly recommended to achieve a high enough quality for industrial use. Furthermore, enzymatic extraction has advantages in controlling the degree of hydrolysis, operating in moderate reaction conditions, and decreasing impurities from salt and waste production [114]. The enzymes derived from plants (bromelain, papain, and ficin), animals (trypsin and pepsin), and proteolytic enzymes, such as collagenase, alcalase, and nutrase, are typically used. Pepsin combined with an acid (acetic or citric acid) is commonly applied among them. Here, using an acid solution could improve the effectivity of the process by distracting the intermolecular salt bonds and Schiff bases and increasing the repulsive charge between the tropocollagen molecules and swell. As a result, the enzyme can effectively digest collagen molecules by breaking the cross-link bonds [133,134,135].

The enzymatic extraction of collagen yields better results than acid extraction alone. However, the obtained collagen characteristics should consider the selection of enzymes because it could affect the features of the collagen; for example, using specific enzymes that remove telopeptides in ASC collagen could result in fiber formation and lower collagen gel strength [136,137]. Telopeptides that contain large amounts of hydrophobic residues greatly assist the inter-microfibrillar interaction. In addition, ASC’s inter- and intra-molecular cross-linking are richer than those of PSC [138]. The presence of telopeptide also affects thermal stability; for instance, the thermal stability between PSC and ASC ranges from 1–4 °C [109,134,139,140,141,142], but in the skin of grass carp collagen, the difference in the denaturation temperature of PSC (telopeptide removed) and ASC (telopeptide retains) is 6.84 °C, with superiority held by collagen that retains telopeptide [143].

#### 3.2.2. Electromagnetic-Assisted Extraction

Other physical electromagnetic interactions in natural product extraction, such as ultrasound and microwave irradiation, have attracted much attention due to their effectiveness in yielding products quickly. Ultrasonic extraction uses ultrasound waves to create cavitation, which allows a physical impulse to generate sufficient kinetic energy to break the cell’s samples during the extraction. This technique is safe, fast, and environmentally friendly. Therefore, it has been widely used in the extraction process, even on an industrial scale.

Ultrasound waves were successfully applied in the extraction of type II collagen from chicken sternal cartilage, using ultrasound with a power of 950 W, 20–25 kHz, amplitude 10, yielding 84.14% (*w*/*w*) at 36 min [144]. The T_d_ value (5 °C) of the obtained collagen was reported to be higher than that of the non-ultrasound-treated product. However, the selection of extraction time should be strictly observed because applying treatments for longer than 36 min caused minor changes to the secondary structure. In addition, Zou et al. [145] reported that the collagen yield obtained in chicken lungs with the ultrasound-assisted method was 31.25% at a power of 150 W. Furthermore, the combination of acid, pepsin, and ultrasound assistance in a meat industry by-product showed an increased yield of pepsin-ultrasound treatment by 36.95% compared to a pepsin-only treatment [146]. In this context, as mentioned earlier, ultrasound treatments do not lead to a significant change in the triple-helix structure of collagen.

In the extraction of marine collagen, the implementation of ultrasound at 20 kHz (80% amplitude) for 24 h was reported to yield an excellent result (~90.40%) from the collagen extraction in sea bass skin [147]. Song et al. [148] developed the ultrasound-assisted method in industrial systems and achieved a two-fold higher yield than the ASC method. Collagen obtained by the ultrasound-assisted method depends on the frequency, amplitude, and extraction time. High frequency, amplitude, and extraction time could produce a higher yield. However, increasing the amplitude and lengthening the extraction time are not always linear [149]. An excess physical force can damage the hydrogen bonds between collagen molecules and break the polypeptide chain bonds [138,147]. Therefore, the use of ultrasound in molecular extraction requires further study.

Like ultrasound, microwaves are intended to provide thermal energy to a sample through microwave radiation, with the capacity to tune the radiation power, thus affecting the amount of heat generated [150]. One study suggests using microwave radiation for 2 min at 25 °C to increase collagen recovery from 51.24% to 76.72% [151]. In addition, Chen et al. [152] reported 1.5 yield enhancement after applying microwave irradiation at 37 °C for 7 hours in an acid solution from bovine skin extraction, without changes in the structure and T_d_ of the obtained collagen. However, the effect seems to have a different result. In the collagen extraction from the bovine Achilles tendon, microwave irradiation showed a slight impact on lowering T_d_ (25 °C) compared with oil bath heating (34 °C) [153]. This result shows that the application of the microwave irradiation method for extracting marine source collagen must be considered since the average denaturation temperature of AQCol is below 30 °C [138].

## 4. Scaffold Fabrication Method

Cross-linking is one of the main treatments that need to be done on collagen prior to its utilization as a scaffold for tissue engineering or other uses, such as drug delivery and wound healing. Cross-linking is a method of fabricating collagen biomaterials that connects collagen fibrils through chemical bonds into fibers as precursor to a fibrous network, which are subsequently suitable as biomaterials, such as sponges, hydrogels, or membranes. The chemical bonds that are formed can improve the physical properties of collagen biomaterials, especially their thermal stability and rheological properties. Several cross-linking methods are available, including chemical cross-linking using 1-ethyl-3-(3-dimethyl aminopropyl) carbodiimide (EDC), which connects the carboxyl and amine groups on the collagen molecule [69,79,154]. Another crosslinking agent, glutaraldehyde, links the aldehyde group with the amine group of lysine [155,156]. Alternatively, there are physical methods for cross-linking collagen biomaterials, namely, dehydrothermal treatment and ultraviolet radiation [157,158]. Dehydrothermal treatment removes the remaining water on the scaffold, causing an intermolecular cross-link between the carboxyl group of glutamate and lysine through a condensation reaction at temperatures above 90 °C under vacuum conditions. In contrast, UV radiation causes a connection between tyrosine and phenylalanine through a radical reaction. Moreover, enzyme-assisted cross-linking using transglutaminase catalyzes the acyl-transfer reaction between the amino group of lysine and the carboxyamide group of glutamine [159]. Enzyme-assisted cross-links are commonly used in tissue engineering applications.

The final product of a collagen scaffold is often obtained using freeze-drying, which is a common method of fabricating a highly porous collagen scaffold with interconnecting pores that form a sponge-like structure (Figure 8) [160,161]. The pores’ structure and size depend on the collagen type, source, and chemical crosslinker used [69,70,72,79]. The pore structure formed results from water removal in the freeze-drying process. The collagen slurry remains frozen until it forms like a crystal, at which point the water contained in the frozen slurry collagen is removed by a sublimation process, by which the water, in the form of ice crystals, immediately becomes steam under vacuum conditions. The water crystals that disappear from the frozen slurry of collagen form a cavity, which will later become a pore on the collagen scaffold [162].

Fibrous collagen scaffolds can be obtained through electrospinning fabrication. An electric potential is applied to the polymer solution to modulate the formation and deposition of polymer fibers [163]. The polymer solution, when given an electric potential, is sprayed through a nozzle towards a grounded target with an opposite charge to the polymer solution reservoir (Figure 8). The solvent evaporates during the fiber transfer process to the grounded target. The size of the fiber formed by electrospinning ranges from nanometers to microns, making it suitable for tissue engineering applications [164,165].

Another approach may involve decellularisation, which requires the use of a natural collagen matrix as a functional biomaterial (Figure 8) [166]. Decellularisation includes removing material and host-cell antigens to prevent immunological rejection, leaving only a water-insoluble protein matrix (mostly collagen) that can support cell attachment and other bioactive agents [167,168]. The procedure includes the isolation of all or part of the tissue and the decellularization and recellularization of the treated collagen. Ott et al. [169] reported on a bioartificial lung fabricated using a lung decellularized by detergent perfusion, which produced a scaffold with acellular blood vessels, airways, and alveoli. The native scaffold implanted epithelial and endothelial cells to support gas exchange and exhibited gas exchange compared to the native lung tissue. However, its application remains challenging concerning the cell recognition of collagen and the consistency of the biomaterial to be applied in vivo [20].

For the use of collagen as a hydrogel, the hydrogelation of collagen is a simple method for preparing collagen as a biomaterial for biological applications (Figure 2). Hydrogelation can be carried out by reacting collagen with a crosslinker agent in a phosphate buffer solution and then leaving it overnight at 37 °C for gelation [71,78]. Another effort to provide collagen hydrogel is shown by Zhang and colleagues. The study highlights that collagen from snakehead fish (*Channa argus*) may form a self-assembly collagen hydrogel. The hydrogel was examined to have an accelerated process affected by the presence of telopeptides compared to atelocollagen [170]. A similar study also shows that the collagen hydrogel from *C. argus* is affected by the precursor concentration, temperature, pH, buffer, and ion strength. In addition, the study also remarks that the correlation of density is more pronounced compared to the fibril size produced in the hydrogelation process [171].

## 5. Tissue Engineering and Regenerative Medicines

### 5.1. Bone Tissue Engineering

Bone is a composite consisting of two major components, an inorganic component of calcium phosphates and an organic element of mainly collagens [172]. Numerous studies have revealed that marine and freshwater-based biomaterial collagens could promote bone tissue regeneration. Some produce minerals within their anatomies, such as the submicron silica frustules of diatoms and the aragonite skeletons of lobster and cuttlefish. Furthermore, the porous nature of marine organisms allows them to be tailored for specific applications to enhance bone ingrowth [173]. The low toxicity of their biomaterial, with unique physical properties that cannot be replicated synthetically, e.g., interconnectivity, pore size distribution, and tortuosity, lead marine organisms to be used either as templates for material design or directly as bone substitute materials.

Several collagens isolated from marine sources, such as jellyfish *S. nomurai meleagris* [102] and sponge *C. reniformis* [174], have been extensively investigated as biomaterials for bone regeneration to replace bovine-derived collagen. The findings are supported by the immunogenicity and zoonosis risk of the bovine’s collagen, particularly of the transmissible spongiform encephalopathies. Hydrogels produced with collagen extracted from the jellyfish and sponge species above have been shown to produce porous structures that support cell attachment (Figure 9) [102]. Collagen from marine sources has been proposed as a safer alternative to this contamination.

A recent in vivo study of collagen hydrolysates (CHs) from freshwater silver carp (*Hypophthalmichthys molitrix*) skin suggested that it could combat osteoporosis in chronologically aged mice. It was observed to have some effects, including increased bone mineral density, Hyp content, and alkaline phosphatase (ALP) level and reduced tartrate-resistant acid phosphatase 5b (TRAP-5b) activity. CHs mainly increased bone remodeling by stimulating the transforming growth factor β1 (TGF-β1)/Smad signaling pathway and improving the interaction between collagen and α2β1 integrin, which indicates the stimulation could apply CHs from fish to alleviate osteoporosis or treat bone loss [175]. Moreover, another in vivo experiment conducted in a rat model with histological analysis showed new bone formation in all groups after eight weeks. Bone formation was significantly increased in treated bone lesions compared to untreated bone tissue. The study showed that the Collgel^®^ product from silver carp skin (*H. molitrix*) can be used instead of conventional porcine or bovine collagen membranes in guided bone regeneration [176]. Despite the necessity for further purification techniques, this collagen has excellent potential in bone tissue engineering applications.

### 5.2. Dental Tissue Engineering

Over the past few decades, a wide range of studies has investigated the provision of tissue-engineered dental grafts, significantly improving the production of scaffolds with similar characteristics to a natural tooth [177]. Since collagen is the most abundant fibrous protein originating from the bone and dentin [178], some efforts have been made to apply marine-derived collagen in dental tissue engineering. Several other materials have been incorporated into collagen to enhance mechanical properties and bone matrix interface strength to form a dental scaffold.

Recently, a salmon collagen (SC) gel from *Oncorhynchus keta* has been investigated to observe its potential application as a scaffold for the in vitro cultivation of human periodontal ligament fibroblasts (HPdLFs). The growth rates, differentiated functions, and morphologies of the cells cultured on SC gel showed higher growth rates and differentiated parts of HPdLFs than on the porcine collagen (PC) gel. The effectiveness of supporting cells to grow efficiently and maintain their differentiated functions exhibited its potency for dental tissue engineering applications [179]. Meanwhile, Tang and Saito [180] investigated the in vitro effects of collagen type I (COL-I) derived from tilapia (*O. niloticus*) scales and porcine skin on a rat odontoblast-like cell line, MDPC-23 in terms of cellular proliferation, differentiation, and mineralization. The results suggest that the tilapia-scale collagen exhibited comparable biocompatibility to porcine skin collagen, as evidenced by increased initial cell attachment, enhanced ALP activity, upregulated gene expression of BSP, and accelerated matrix mineralization.

Another study on dental tissue engineering recently developed tilapia collagen augmented by an appropriate bioactive glass precursor solution and chitosan to develop biomimetic fish collagen/bioactive glass/chitosan (Col/BG/CS) composite nanofiber membranes. The composite membrane showed some antibacterial activity against *Streptococcus mutans* and could promote the adhesion, viability, and osteogenic differentiation of human periodontal ligament cells. An in vivo study showed that the Col/BG/CS membrane could promote bone regeneration in the furcation defect of dogs, indicating that a biomimetic fish Col/BG/CS composite has the potential membrane to induce periodontal tissue regeneration with a certain degree of antibacterial activity [181].

### 5.3. Cartilage Repair

Cartilage is an avascular tissue and articular skeletal tissue consisting of cells in ECMs. Its mineralization depends on the cartilage type, which is deposited by cartilage-forming cells, such as chondroblasts and chondrocytes, and is removed by mono- and multinucleated chondroclasts. The cartilage has a minimal ability to self-regenerate due to its avascular structure. Avascular cartilage defects heal poorly, leading to catastrophic degenerative arthritis [182]. Thus, the application of articular cartilage tissue engineering to repair, regenerate, and improve injured articular cartilage needs to be studied further [54]. Cartilage is primarily composed of collagen type II; thus, the use of collagen type II scaffolds has been advocated in cartilage engineering approaches [183].

A previous study evaluated the biocompatibility of marine collagen harvested from jellyfish *R. esculentum*. The study analyzed cell migration, cytotoxicity, and extracellular matrix formation using human and rat nasal septal chondrocytes [184]. Seeded and unseeded scaffolds were transplanted into nasal septum defects in an orthotopic rat model. The surgical procedure is a suitable way to evaluate new scaffold materials for their applicability in the context of nasal cartilage repair. Overall, the in vivo results indicated that marine collagen obtained from the jellyfish is a promising new scaffold for nasal cartilage tissue engineering that presents no cytotoxic effects. At the same time, marine collagen offers excellent biocompatibility, with only slight evidence of local inflammatory reactions [184]. Scaffolds are suitable for effectively preventing nasal septal perforations, especially when seeded with autologous chondrocytes.

In line with the previous approach, hybrid scaffolds based on marine biomaterials containing soft hydrogel alginate combined with a mechanically stiffer and porous collagen scaffold from *R. esculentum* were evaluated to mimic both main tissue components of cartilage. The hybrid constructs of jellyfish collagen and alginate supported human mesenchymal stem cells’ chondrogenic differentiation and provided more stable constructs than pure hydrogels, indicating their suitability for biomedical articular cartilage therapies [185]. Another study showed that all marine collagens (jellyfish/shark)-chitosan-fucoidan hydrogel formulations provide an excellent structural architecture and microenvironment, highlighting the usefulness of H_3_ biomaterials due to the large number of polymers in their composition [186]. The results suggest that all the developed marine hydrogels are promising as template devices for tridimensional cell cultures. Thus, the results support the proliferation of chondrocyte-like cells, resulting in more biomechanically stable hydrogels that are interesting for future applications in cartilage tissue engineering and regenerative medicine.

### 5.4. Vascular Tissue Engineering

Marine collagens have been shown to represent an alternative scaffold material for vascular tissue engineering and bioengineered tissues, such as blood vessels and heart valves. Novel tubular scaffolds composed of a porous collagen matrix and a fibrous poly(lactide-co-glycolide) (PLGA) layer, which were fabricated by freeze-drying and electrospinning processes for vascular grafts, improved the mechanical strength of collagen under a pulsatile perfusion system on the proliferation and phenotype of smooth muscle cells and endothelial cells cultured on collagen/PLGA scaffolds [187]. Meanwhile, a recent in vivo animal mouse model study of collagen from European carp (*Cyprinus carpio*) assessed the systemic and local immunological response to subcutaneous implants of a vascular graft covered with collagen. The results indicated that the implants covered with carp collagen induce an immunological response comparable to that of bovine collagen-covered implants in a mouse model [188]. Another in vivo study of crosslinked collagen from young (≤2 kg) scaleless carp evaluated a sheep model vascular graft’s mechanical and immunological properties [189]. The results showed that modification of the fat content of collagen used in the outer and inner layer of a composite three-layer vascular graft plays a novel critical role in its potency and structural changes in the preclinical sheep model.

Furthermore, an in vitro study of CHs from silver carp (*H. molitrix*) skin assessed the bioavailability of bioactive peptides by the Caco-2 cells model [190]. This study indicated that the oligopeptides GPR, GPRG, and GPRGP are a potential index of bioactive compounds in the preparation of anti-thrombotic functional foods and health care supplements for people at the pre-thrombotic state for the application of cardiovascular disease.

### 5.5. Wound Repair and Skin Tissue Engineering

The biomedical applications of aquatic-based collagen are not limited to bone and cartilage, as they also encompass wound repair and skin tissue engineering [191]. Skin tissue engineering using scaffolds overcomes the limitations of several wound healing processes, such as autografts, allografts, and xenografts. The success of tissue engineering lies in the optimal integration of cells, biomaterial scaffolds, and cellular signals, such as growth factors. A critical step of all tissue engineering techniques is the construction of a 3D scaffold mimicking the ECM, which serves as physical support to guide the initial cell attachment and subsequent tissue formation [192].

The skin tissue engineering of marine collagen scaffolds from jellyfish *Cassiopea andromeda* was analyzed in vitro using human dermic fibroblasts. The results demonstrated that the decellularization process reduces the native cell population, leading to a 70% reduction in DNA content. Moreover, the cytotoxicity test indicated that the decellularized scaffolds are suitable for cell adhesion and the proliferation of human fibroblasts. Therefore, the results suggested that the cell-free scaffold could be used in skin tissue engineering [193]. Another experiment conducting *Paralichthys olivaceus* skin collagen/alginate (FCA) sponge scaffold with in vitro test in neonatal human dermal fibroblasts (NHDF) exhibited the best cellular compatibility. The results suggested that the number of NHDF-neo cells was always the highest in the FCA/COS1 scaffold compared to other scaffolds, emphasizing that the FCA/COS1 scaffold is an excellent candidate for cell adhesion and proliferation in skin tissue engineering applications [187].

The wound healing process consists of four significant stages: hemostasis, inflammation, proliferation, and tissue remodeling. During the inflammatory phase, chemotactic signals recruit specialized cells to clean the injury from foreign materials. A fibrin eschar is formed before a granulation tissue is introduced by extracellular matrix deposition, including collagen [194]. Collagen plays a vital role in every step of wound healing, as it attracts cells, such as keratinocytes and fibroblasts, to the wound area, enhancing debridement, angiogenesis, and re-epithelialization [195]. 

A recent clinical study compared the wound healing effect of a commercial marine collagen fish skin graft (Kerecis^®^) with a bovine collagen skin graft (ProHeal^®^). The skin graft donor sites treated with the Kerecis^®^ treatment healed significantly faster (approximately two days earlier) than non-treated wounds and those treated with ProHeal^®^ bovine collagen. The average wound healing time for Kerecis^®^ was 10.7 ± 1.5 days, and that for ProHeal^®^ was 13.1 ± 1.4 days. The faster healing of the Kerecis^®^ treatment, compared to that of the ProHeal^®^ treatment, might be due to the synergistic effect of the unique biophysical structure and the bioactive components of acellular fish skin. Additionally, the cell proliferation on Kerecis^®^ was much better than on ProHeal^®^ on Day 1 and Day 3 of culture using human dermal fibroblasts, human epidermal keratinocytes, and human umbilical vein endothelial cells. This outcome indicates that Kerecis^®^ has a higher capability of cell proliferation and, therefore, skin regeneration than the ProHeal^®^ substrate. Based on the results, Kerecis^®^ can be considered a viable alternative to the bovine collagen graft, with high therapeutic potential for extensive burn wound management [196].

In a preclinical wound healing study using a rat model, various biomaterials, including PSC, ACS from Tilapia, and bovine collagen sponge (BCS), showed that the PSC and ACS groups revealed significantly better wound healing ability than control groups and a slightly better wound healing ability than the BCS group. The results were based on the macroscopic wound observation and the Hyp component of the collagen, which served as an indicator to determine collagen deposition and measure wound healing speed. On day 14 after wound incision, the PSC- and ACS-treated groups had higher Hyp content (7.32 ± 0.43 mg/g, 7.41± 0.42 mg/g tissue) than the control and woundplast-treated groups. This result might be due to the three-dimensional features and higher porosity of PSC and ACS than BCS. This study showed that the PSC- and ACS-treated rats exhibited accelerated wound healing in the rat model [35].

A previous study suggested that commercial hydrolyzed collagen product (Vinh Hoan Corporation) from freshwater fish *Pangasius hypophthalmus* supplemented with commercial Vinh Wellness Collagen powder was safe and well-tolerated clinically. The results of this study support the use of fish-derived hydrolyzed collagen to improve skin health in an aging population [197]. Another commercial product, Geistlich Bio-Gide^®^ membrane, was used in an in vitro study of cytotoxicity on human fibroblast of the skin of silver carp. The research revealed that 1% of fish collagen affected the proliferation of human fibroblasts compared to the control group, without producing any significant cytotoxic effects. This result indicates that the experimental collagen is a safe material for healing wounds and reducing scar tissue [198].

Based on earlier findings, collagen extracted with the acidic method from the skin of silver carp (*H. molitrix*) indicates the skin reconstructive treatment potential of acidic collagen extracts acquired from the skin of silver carp containing considerable amounts of small 7–29 peptides. Therefore, the application of these peptides could result in beneficial clinical effects in patients needing reconstructive treatment [199]. A fish market in Fuzhou, China, supplied another useful silver carp fish skin to form freshwater-derived collagen in the form of collagen/chitosan/chondroitin sulfate scaffolds. The in vitro and in vivo results suggested that the scaffolds had good biocompatibility and could promote fibroblast cell proliferation and skin tissue regeneration. The results also demonstrated that this scaffold-controlled release system could be used in skin tissue engineering [200]. 

## 6. Future Perspective

Collagen is widespread throughout the human body and possesses excellent biological properties for the regeneration of almost every organ in the human body. These properties include excellent absorbability in the body, low antigenicity, non-toxicity, high tensile strength, high affinity with water, good biocompatibility, and biodegradability. This aqueous-based colloidal gel also can load cells and bioactive compounds. Their low immunogenicity diminishes the likelihood of rejection when ingested or injected to a foreign body. Current commercial collagens are mainly derived from bovine and porcine. However, the increasing market demand for collagens and some barriers regarding religious and potential zoonic risk have encouraged the exploration of alternative natural resources. 

AQCols are potential alternative resources to produce collagens. Their resources are abundant and scalable because they can be obtained from the cultivation of aquatic organisms. These biomaterials could also be obtained from seafood by-products. Utilizing these by-products is beneficial not only because it reduces the waste produced by the fisheries industries, but also because it increases the economic value of this low-value by-product. Their usage minimizes the risks of zoonotic diseases and bypasses most religious restrictions. AQCols have also been presented to have good biocompatibility by not exhibiting severe immunogenic responses.

Despite their potential applications, there are some issues regarding the large exploitation of AQCols. Generally, native AQCols have different amino acid compositions, which generally provide lower thermal stability than BPCols. Marine collagens, especially those from cold-water organisms, have a low denaturation temperature, which limits their processing condition with the extensive use of cold rooms. In addition, collagen extraction should be done immediately after the harvesting of marine tissues, as the organic portion of marine waste can rapidly decompose within hours in the absence of preservation controls [14,201]. The decomposition is mainly caused by bacteria, enzymatic autolysis, and lipid oxidation, and it affects the integrity of the collagen and introduces the risk of bacterial endotoxin contamination. Therefore, the preservation of marine waste before collagen extraction is one of the major steps in the large-scale utilization of AQCols. Moreover, the challenge of utilizing AQCols for biomedical applications is related to the fact that harvesting marine organisms can have direct effects on ecosystems, thus reducing native populations, as well as indirect effects, such as depleting fish feedstocks [31,202]. The large-scale use of non-cultivated marine organisms could be limited by the sustainability feedstock. Therefore, aquatic preservation should always be taken into consideration when selecting marine organisms for commercial tissue engineering applications [14], and maintaining the supply of raw materials is critical in the commercialization of AQCols.

Their chemical structure must be modified to improve their physico-chemical characteristics in some cases. The fabrication of modified hybrids has been reported to be beneficial in improving their thermal stability. This could be done by several methods, such as crosslinking, blending, or chemical modifications. Additionally, explorations of collagens derived from various warm-water environments with high thermal stability (as opposed to cold-water marine species) might be another approach since there are still many underexplored areas. In addition, one crucial factor in tissue engineering is the maintenance of pH and ionic strength as the key factors in achieving linear viscoelasticity properties and transparency [203]. Moreover, mimicking the specific morphology of cell-laden collagen hydrogel is necessary. Thus, to increase the bioactivity and function of hydrogels, such as native tissue, bioactive compounds and stem cells can be added during the collagen hydrogel fabrication process [203,204]. Therefore, the use of marine collagen as a biomaterial in tissue engineering is possible after cross-linking treatment [102].

Based on in vitro studies, seeding cells on the encapsulated biomaterials after the fabrication process could be a time-consuming and challenging way to achieve full cellularization. This is mainly because of the harsh conditions, such as electrospinning, for crosslinking treatment to improve collagen fibrillogenesis. Considering this, more fabrication methods suitable for cell survival in vitro need to be discovered. To date, numerous cross-linking methods have been employed to stabilize collagen, as it can be divided into physical and chemical treatments. Some physical treatments, such as UV irradiation, gamma irradiation, and dehydrothermal treatments, have been developed, as have several chemical treatments, for instance, the use of glutaraldehyde, carbodiimide, or EDC. Furthermore, chemical treatments confer remarkably high strength and stability to the collagen matrix, although they can result in potential cytotoxicity or poor biocompatibility, whereas physical treatments provide sufficient stability with no cytotoxicity [54]. 

Furthermore, various in vivo tissue regeneration processes require diverse degradation rates of collagen-based biomaterials. One of the common methods for controlling the degradation rate is changing the crosslinking degree and the proportion of biomaterial composites. Thus, more research should focus on the mechanism of collagen degradation in vivo at different regeneration stages for tissue engineering applications [205].

The use of marine collagen in tissue engineering applications is still in its infancy. More research in this area would be necessary to gain a deeper exploration of marine collagen scaffolds’ fabrication with various methods to test the mechanical properties, internal pore size, cell encapsulation, and degradation rate for specific applications, such as cartilage, skin, or bone repair. For cartilage repair, the use of marine collagen supported the three-dimensional chondrocyte-like cell culture proliferation, but more testing is needed to verify the phenotype of the cells by using primary chondrocyte cells [186].

Marine collagen could delay and protect against osteoporosis and osteoarthritis because it is capable of increasing bone mineral density, mineral deposition, and osteoblast maturation and proliferation for bone tissue engineering application. However, more studies are needed to report the collagen fiber organization pre- and post-successful mineralization scaffold more accurately. The utilization of innovative 3D-printing techniques that have the potential to create both ideal microenvironments of aligned and densely packed collagen fibers, as well as complex and tailorable 3D architectures, presents the next step forward in the mineralization of bone tissues [206].

For skin tissue engineering, marine collagen demonstrated wound healing both in vitro and in vivo, specifically promoting skin re-epithelization, vascularization, fibroblast migration, and faster wound healing rates [191]. Marine collagen is also responsible for increasing the viability of periodontal ligament cells. The tilapia collagen scaffold has the ability to enhance periodontal tissue repair, and it exhibits anti-bacterial properties and increases cell viability, cell adhesion, and osteogenic expression in periodontal ligament cell [207].

As the implementation of marine collagen in tissue engineering is still an emerging research topic, most recent studies have been carried out in vitro, with few clinical studies evaluating aspects related to safety and efficacy. While marine tissues lack mammalian antigens, e.g., galactose-α-1,3-galactose, that cause allergies to mammalian tissues, and although quality control measures can verify the removal of allergens after collagen extraction, certain marine species may present allergens that can trigger allergic reactions in humans [208]. Thus, it is important to examine the patient’s medical history for seafood allergy to prevent unexpected allergic reactions.

Another emerging study is collagen-based bioink for different tissue engineering applications and regenerative medicines, specifically the combination of collagen-based bioink and 3D bioprinting [209]. However, the applicability of collagen for 3D bioprinting depends on the BPCols source, with a very limited study involving AQCols and only a few commercially concentrated collagen bioinks available. One of the distinctive characteristics of these commercial bioinks is the possibility to add cells and any components of the extracellular matrix to their composition to produce the bioprint of an artificial cell-laden matrix. Therefore, more studies are required to explore the behavior of cells in 3D-bioprinting AQCols, including their proliferation, migration, differentiation, functionality, and retention, as well as the density of the collagen for biomedical and regenerative medicine application.

## Figures and Tables

**Figure 1 marinedrugs-21-00087-f001:**
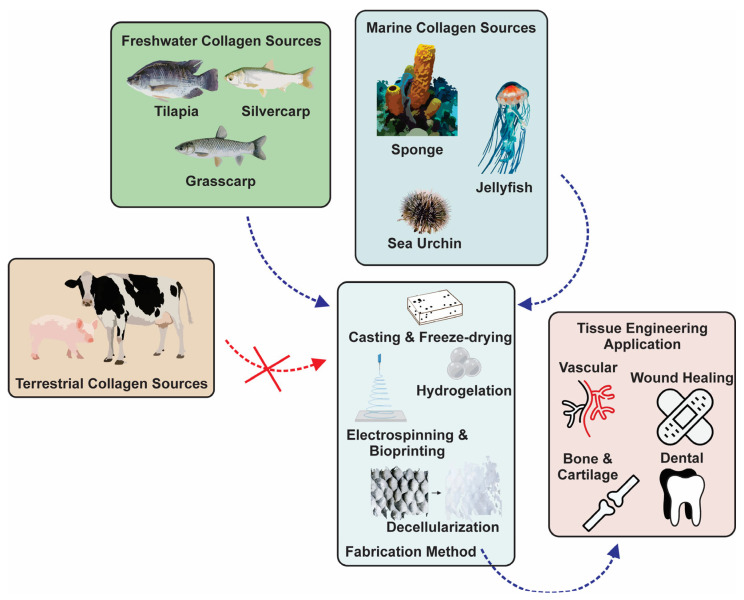
Various sources, fabrication methods, and tissue engineering applications of aquatic collagen.

**Figure 2 marinedrugs-21-00087-f002:**
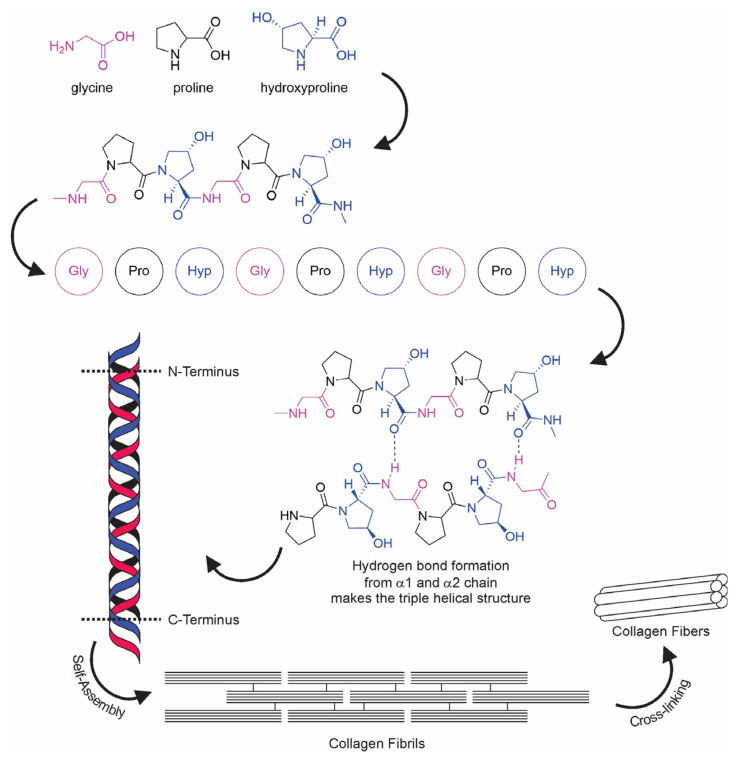
Collagen fiber formation.

**Figure 3 marinedrugs-21-00087-f003:**
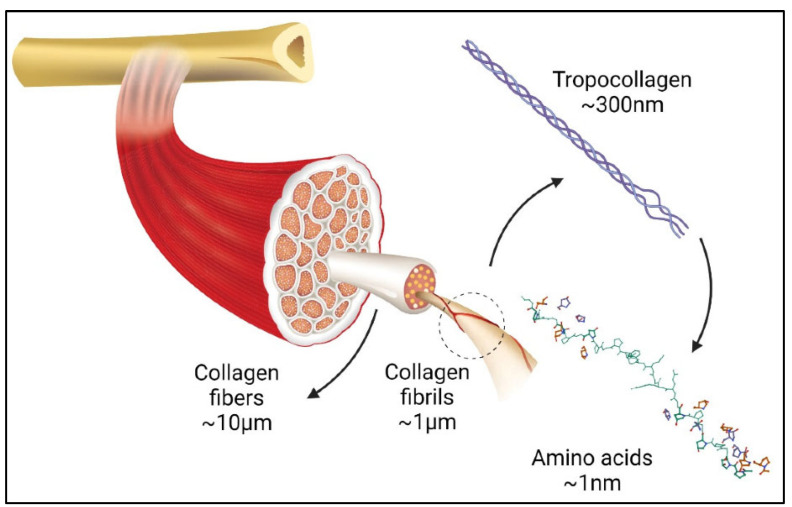
Graphic illustration of collagen in organ tissues adapted from [44].

**Figure 4 marinedrugs-21-00087-f004:**
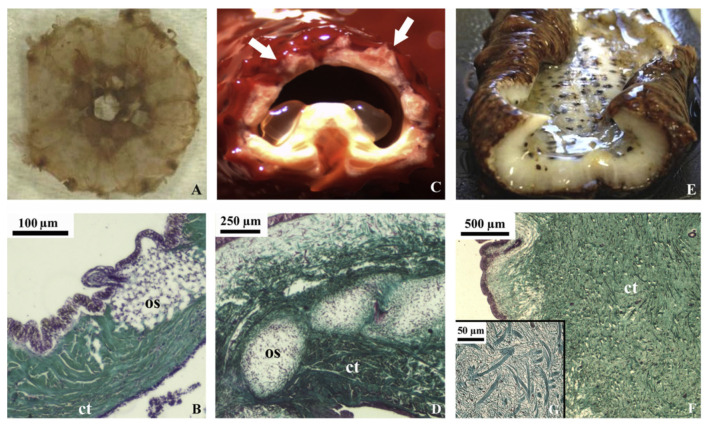
Echinoderm tissues for the collagen extraction protocol in in vivo (stereomicroscopy) and histological sections (light microscopy, Trichrome Milligan staining, collagen is stained in green). (**A**) sea urchin peristomial membrane (PM); (**B**) PM connective tissue characterized by highly packed collagen fibrils and fibers (ct) with quite large ossicles (os); (**C**) starfish aboral arm wall (AW; arrows); (**D**) AW connective tissue characterized by dense collagen fiber bundles (ct) in which large ossicles (os) are widespread; (**E**) sea cucumber body wall (BW); (**F**) BW connective tissue with small spicules and the absence of highly packed collagen fibers (ct); (**G**) details of F on loosely packed collagen fibrils (these figures are adapted with permission from Elsevier) [66].

**Figure 5 marinedrugs-21-00087-f005:**
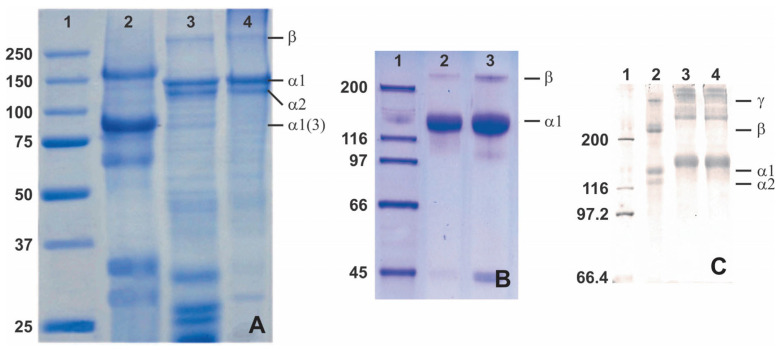
SDS-PAGE analysis of collagen from various sea cucumbers: (**A**)—lane 1 (protein marker), lane 2 (body wall of a sea cucumber *Holothuria cinerascens*), lane 3 (tilapia fish skin), and lane 4 (porcine skin), adapted with permission from [90]; (**B**)—lane 1 (protein marker), lane 2 (skin *Parastichopus californicus*), and lane 3 (connective tissue *P. californicus*), adapted with permission from [60], Copyright 2010 American Chemical Society; (**C**)—lane 1 (protein marker), lane 2 (calf skin collagen), lane 3 (*S. monotuberculatus* collagen without β-mercaptoethanol), and lane 4 (*S. monotuberculatus* collagen with β-mercaptoethanol), adapted with permission from John Wiley and Sons [51].

**Figure 6 marinedrugs-21-00087-f006:**
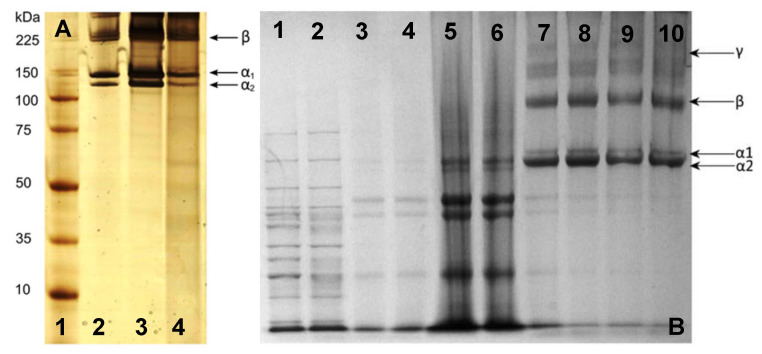
SDS-PAGE analysis of collagen from jellyfish *C. mosaicus*: (**A**) [77]—lane 1 (high-molecular-weight protein marker), lane 2 (type I collagen from rat tail), lane 3 (extracted collagen from rat tail tendon), and lane 4 (extracted collagen from jellyfish oral arm) (Reprinted (adapted) with permission from Rastian, Z.; Pütz, S.; Wang, Y.; Kumar, S.; Fleissner, F.; Weidner, T.; Parekh, S.H. Type I Collagen from Jellyfish Catostylus mosaicus for Biomaterial Applications. ACS Biomater. Sci. Eng. 2018, 4, 2115–2125, doi:10.1021/acsbiomaterials.7b00979. Copyright 2018 American Chemical Society); and jellyfish *Rhizostoma pulmovarious*: (**B**) Adapted with permission from [36]—lanes 1–2 (high-molecular-weight protein marker), lanes 3–6 (noncollagenous proteins removed from solution prior to testing), and lanes 7–10 (collagen solution containing α1, α2, β, and γ collagen chains).

**Figure 7 marinedrugs-21-00087-f007:**
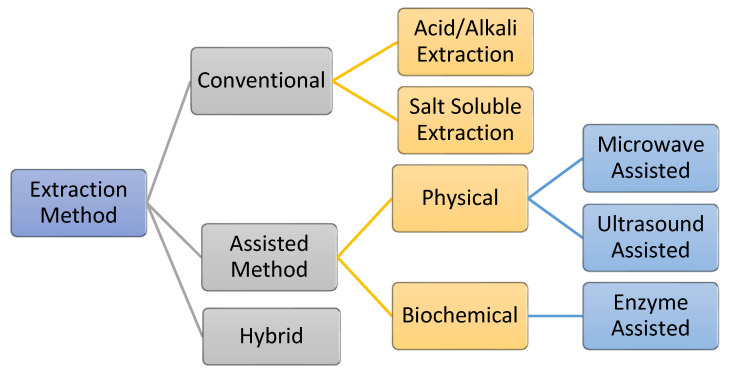
Various collagen extraction methods, including a hybrid method that combines conventional and assisted methods.

**Figure 8 marinedrugs-21-00087-f008:**
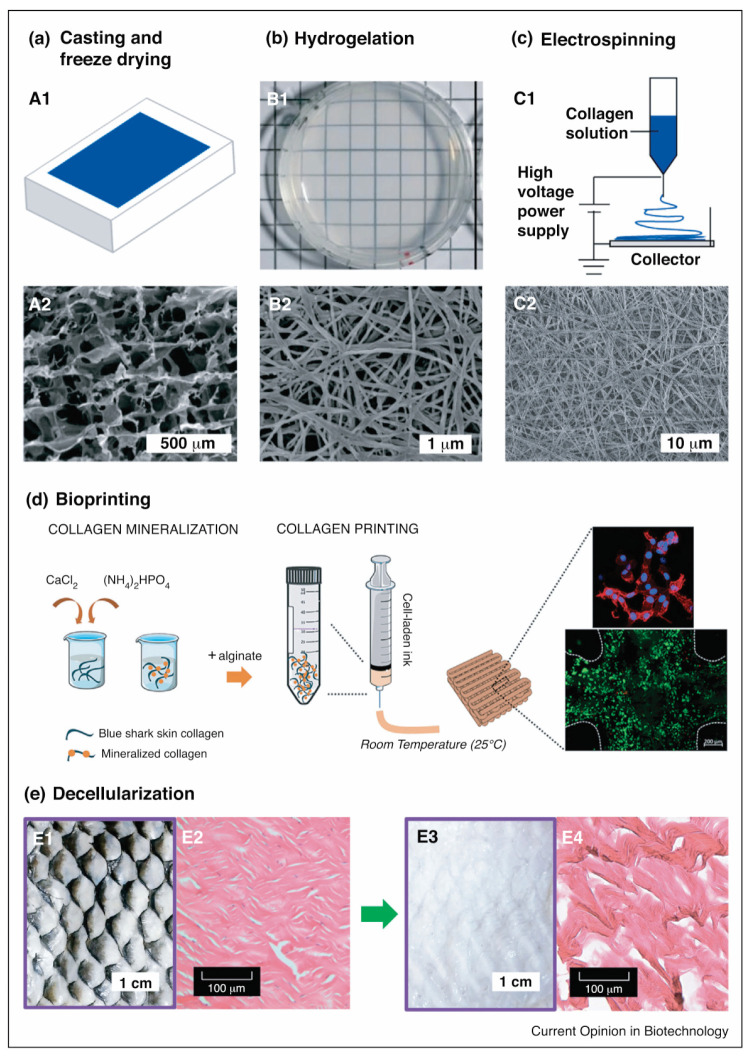
Marine collagen scaffold fabrication techniques. (**a**) Casting and freeze-drying–(A1) collagen sponge via casting on a mold followed by freeze-drying, and (A2) SEM image of a casted collagen sponge; (**b**) Hydrogel fabrication via self-assembling–(B1) Bright-field image of tilapia collagen hydrogel, and (B2) SEM image of tilapia collagen hydrogel showing a nanofibrillar architecture; (**c**) Electrospinning nanofiber fabrication–(C1) Scheme of electrospinning process, and (C2) SEM image of electrospun tilapia skin collagen; (**d**) Schematic representation of the fabricating cell-laden mineralized collagen/alginate hydrogels using bioprinting dispensing systems, in which cells were spreading (stained red with phalloidin) and live (stained green with calcein AM); and (**e**) Decellularization methods. Bright-field microscope image (E1, E3) and H&E-stained sagittal sections (E2, E4) of tilapia skin (E1, E3) and decellularized tilapia skin (E2, E4), in which extensive removal of pigments and DNA while retaining collagen fibers were observed in decellularization skin. (This set of figures is adapted with permission from Elsevier) [14].

**Figure 9 marinedrugs-21-00087-f009:**
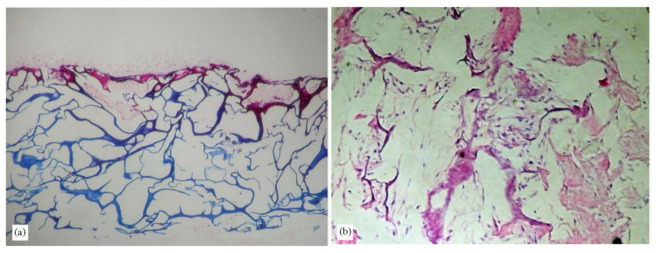
Photomicroscopy images of porous jellyfish scaffolds cultured with fibroblasts for 14 days. Scaffolds were histologically stained with H&E; (**a**) 100× and (**b**) 200× [102] (adapted with permission from Elsevier).

## Data Availability

Not applicable.

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
