# Peer review of "Cutting Edge Aquatic-Based Collagens in Tissue Engineering"

_marinedrugs, 2023, doi:10.3390/md21020087_

Round 1

Reviewer 1 Report

This paper reports an overview of the characteristics of different aquatic collagens, including their extraction and fabrication. This manuscript may be recommended for publication in Marine Drugs after minor revision indicated below.

 - In the Future Perspectives Section, include specific applications for the different collagen to be used as raw materials.

- Include more references of the journal.

-  Include more recent references (from the last 3 years).

Author Response

Responses to Reviewer 1
Manuscript ID Number: marinedrugs-2089531
Manuscript Title: Cutting Edge Aquatic-based Collagens in Tissue Engineering
We thank the referees for their time spent carefully reviewing the manuscript. Below are our responses to the comments of reviewers. We have copied and pasted their comments. The reviewer’s as well as editor comments appear in italic and follow in the same order as written by the referee. Our responses appear in regular font.

Comment
In the Future Perspectives Section, include specific applications for the different collagen to be used as raw materials.
Response
We agree with the reviewer that including specific applications in the future perspectives section would be useful to demonstrate the marine collagen future studies in the skin, bone, and cartilage tissue engineering applications. So, we already added these various applications. 
We thank the reviewer for pointing this out. 

Comment
Include more references of the journal
Response
We appreciate the reviewer’s insightful suggestion. So, have updated more references to the  paper, 

Comment
Include more recent references (from the last 3 years)
Response
We have also updated more recent references, specifically from the last three years’ publications, particularly in the future perspectives section

Reviewer 2 Report

This paper focused on the overview the characteristics, extraction methods and properties of aquatic-based collagens, and their potential application. This is an interesting topic. However, it is suggested that the content of this paper should be further optimized and the language should be further simplified to focus on the research results and the corresponding implications of aquatic collagen. The introduction portion is too long. The characteristics and regularity of aquatic collagen and the optimal path of its application in the field of tissue engineering are not fully described. Some other questions are also listed as following:

1. “2. Aquatic-based Collagen” portion: There are much literatures about fresh-water fishes collagen and marine fishes collagen, however, there are few descriptions of fish collagen in this portion. Why? Why is it highlighted Sea Cucumber Collagen, Sea Urchin Collagen, Jellyfish Collagen, Marine Sponge Collagen in this article?

Such as Journal of Aquatic Food Product Technology, 2014, 23, 264-277; Journal of food, agriculture & environment, 2013, 11, 394-399.

2. “4.1. Conventional Method” portion: What is the definition of Conventional Method about extraction and isolation of AQCol. The acid-soluble collagen (ASC) and pepsin-soluble collagen (PSC) always extracted with acid solution without or with enzymes, respectively, and both are common methods.

3. “4. Scaffold Fabrication Method” portion: self-assembly is also a common method to fabricate the collagen based scaffolds, so the description about the self-assembly of AQCols should be added in this review. Such as Food Biophysics, 2016, 11, 380-387; Macromolecular Research, 2017, 25, 1105-1114.

4. Line 72, The description about the synthetic polymers might be removed as it is not relevant to the topic of this manuscript.

5. Line 88 “while other EMCs are extracted from proteins, e.g., collagen, gelatin, elastin, fibrin, and polysaccharides, e.g., alginate, agarose, chitosan, dextran, hyaluronic acid.” This sentence is not clear.

6. Line 105, what is the nano- and micron-sized fibrous collagen?

7. Line 112, “BPCols have challenges related to their complex tunable mechanical characteristics, sensitivity to enzyme degradation, and low thermal stability [27–30].” Here, the author claimed that the BPCols have low thermal stability. How about the aquatic-based collagen?

8. Line 143, the description about the collagen might be removed to the introduction portion.

9. Line 171, “A recent study suggested that the amino acid content of Pro and Hyp extracted collagen from warm-water fish is more durable in a high-temperature environment than in a low-temperature environment.” This sentence is not clear.

10. Line 196. “Sea cucumber’s collagens are a type I and consist of three α1 chains with a triple helical structure [61].” As well known to all, the type I collagen is composed of two α1 chains and one α2 chain. In ref. 61, the band of α2 chains might overlapped with the band of α1 chains in the SDS-PAGE image. Much more references should be compared, and the consensus conclusions should be cited in the reviews.

11. Line 462. “Using specific enzymes that remove telopeptides in ASC collagen could result in fiber formation” This sentence is not clear.

12. Line 467. “for instance, thermal stability between PSC and ASC ranges from 1–4 °C [109,134,139–142], ” This sentence is not clear.

13. Line 515. “Cross-linking is a method of fabricating collagen biomaterials that connects collagen fibrils through chemical bonds into fibers” According to our research, the cross-linking of collagen might result in the fibrious network.

Author Response

Responses to Reviewer 2

Manuscript ID Number: marinedrugs-2089531

Manuscript Title: Cutting Edge Aquatic-based Collagens in Tissue Engineering

We thank the referees for their time spent carefully reviewing the manuscript. Below are our responses to the comments of reviewers. We have copied and pasted their comments. The reviewer’s as well as editor comments appear in italic and follow in the same order as written by the referee. Our responses appear in regular font.

Comment:

“2. Aquatic-based Collagen” portion: There are much literatures about fresh-water fishes collagen and marine fishes collagen, however, there are few descriptions of fish collagen in this portion. Why? Why is it highlighted Sea Cucumber Collagen, Sea Urchin Collagen, Jellyfish Collagen, Marine Sponge Collagen in this article?

Such as Journal of Aquatic Food Product Technology, 2014, 23, 264-277; Journal of food, agriculture & environment, 2013, 11, 394-399.

Response:

Thank you for your remark, the description of fish collagen (marine and freshwater) is highlighted in section “3.3 Fish Collagen” portion. We try to emphasize the other sources of collagen to provide a wider range of potential tissue engineering application. In addition, the description on fish collagen also lies on a few parts that focus on its application and on the extraction method, i.e.,

Line 506-510

The presence of telopeptide also affects thermal stability; for instance, thermal stability between PSC and ASC ranges from 1–4 °C [109,134,139–142], but in the skin of grass carp collagen, the difference in denaturation temperature of PSC (telopeptide removed) and ASC (telopeptide retains) is 6.84 °C with superiority held by collagen that retains telopeptide [143].

Line 664-666

A recent in vivo study of collagen hydrolysates (CHs) from freshwater silver carp (Hypophthalmichthys molitrix) skin suggested that it could combat osteoporosis in chronologically aged mice.

Line 674-676

The study showed that the Collgel® product from silver carp skin (H. molitrix) can be used instead of conventional porcine or bovine collagen membranes in guided bone regeneration [176].

Line 823-825

A previous study suggested that commercial hydrolyzed collagen product (Vinh Hoan Corporation) from freshwater fish Pangasius hypophthalmus supplemented with commercial Vinh Wellness Collagen powder was safe and well-tolerated clinically.

Line 837-839

A fish market in Fuzhou, China, supplied another useful silver carp fish skin to form freshwater-derived collagen in the form of collagen/chitosan/chondroitin sulfate scaffolds.

Comment:

“4.1. Conventional Method” portion: What is the definition of Conventional Method about extraction and isolation of AQCol. The acid-soluble collagen (ASC) and pepsin-soluble collagen (PSC) always extracted with acid solution without or with enzymes, respectively, and both are common methods.

Response:

Thank you for your remark, the mentioned conventional method reffered to the absence of extraction asisstance such as microwave, ultrasound, or enzyme.

In the case of possible misunderstanding, we only include acid or alkali treatment and salt soluble extraction process which does not invole enzyme assistance in the extraction process. In addition, the last paragraph that mention PSC is for comparison purpose to the discussed salt-soluble collagen in terms of yield and physical characteristic (denaturation and melting temperature) affected by extraction method.

Comment:

“4. Scaffold Fabrication Method” portion: self-assembly is also a common method to fabricate the collagen-based scaffolds, so the description about the self-assembly of AQCols should be added in this review. Such as Food Biophysics, 2016, 11, 380-387; Macromolecular Research, 2017, 25, 1105-1114.

Response:

Thank you for your remarks, the self-assembly is part is included in the section of Scaffold Fabrication Method (Line 629-635)

Comment:

Line 72, The description about the synthetic polymers might be removed as it is not relevant to the topic of this manuscript.

Response:

Thank you for your remarks, the paragraph is removed

Comment:

Line 88 “while other EMCs are extracted from proteins, e.g., collagen, gelatin, elastin, fibrin, and polysaccharides, e.g., alginate, agarose, chitosan, dextran, hyaluronic acid.” This sentence is not clear.

Response:

Thank you for your remarks, the sentence is included in the removed paragraph

Comment:

Line 105, what is the nano- and micron-sized fibrous collagen

Response:

Thank you for your remarks, nanometer and micrometer-sized describes the different size of collagen produced by scaffold fabrication method such as electrospinning that may involve the fibril diameter to be adjusted.

Pham, Q.P.; Sharma, U.; Mikos, A.G. Electrospinning of Polymeric Nanofibers for Tissue Engineering Applications: A Review. Tissue Eng. 2006, 12, 1197–1211, doi:10.1089/ten.2006.12.1197.

Subbiah, T.; Bhat, G.S.; Tock, R.W.; Parameswaran, S.; Ramkumar, S.S. Electrospinning of nanofibers. J. Appl. Polym. Sci. 2005, 96, 557–569, doi:10.1002/app.21481.

Comment:

Line 112, “BPCols have challenges related to their complex tunable mechanical characteristics, sensitivity to enzyme degradation, and low thermal stability [27–30].” Here, the author claimed that the BPCols have low thermal stability. How about the aquatic-based collagen?

Response:

Thank you for your remarks, the aquatic-based collagen properties are emphasized in the section Aquatic-based collagen, in particular line 191-208. The paragraph also served as the background of the importance on fabrication method to improve physical properties of the produced collagen scaffolds. Although some may have different findings but fabrication method provides the ability to tune physical properties to serve a specific tissue engineering application.

Comment:

Line 143, the description about the collagen might be removed to the introduction portion.

Response:

Thank you for your remarks, the description of collagen is unable to be moved to the introduction due to it is serve as the background on emphasizing AQCols on the last two paragraph of the section.

Comment:

Line 171, “A recent study suggested that the amino acid content of Pro and Hyp extracted collagen from warm-water fish is more durable in a high-temperature environment than in a low-temperature environment.” This sentence is not clear.

Response:

Thank you for your remarks, the sentence is now fixed as follows,

A recent study suggested that the amino acid content of Pro and Hyp extracted collagen from fish with warmer habitat (6 species – Pterocaesio digramma, Coryphaena hippurus, Pseudorhombus pentophthalmus, Pleuronichthys cornutus, Scomber australasicus, and Decapterus tabl) is more durable compared to collagen produced from colder habitat (5 species – Pleurogrammus azonus, Synaphobranchus bathybius, Coryphaenoides pectoralis, Coryphaenoides acrolepis, and Lycenchelys squamosal). The study also examined the presence of another amino acid, serine (Ser), which increases the flexibility of its fibrous collagen.

Comment:

Line 196. “Sea cucumber’s collagens are a type I and consist of three α1 chains with a triple helical structure [61].” As well known to all, the type I collagen is composed of two α1 chains and one α2 chain. In ref. 61, the band of α2 chains might overlapped with the band of α1 chains in the SDS-PAGE image. Much more references should be compared, and the consensus conclusions should be cited in the reviews.

Response:

Thank you for your remarks, the consensus has been written on the sentence with addition of references

Comment:

Line 462. “Using specific enzymes that remove telopeptides in ASC collagen could result in fiber formation” This sentence is not clear.

Response:

Thank you for your remarks, the sentence now combined with previous sentence to provide clarity

Line 501-504

However, the obtained collagen characteristics should consider the selection of enzymes because it could affect the features of the collagen, for example using specific enzymes that remove telopeptides in ASC collagen could result in fiber formation and lower collagen gel strength

Comment:

Line 467. “for instance, thermal stability between PSC and ASC ranges from 1–4 °C [109,134,139–142], ” This sentence is not clear.

Response:

Thank you for your remarks, the sentence is now clear by addition of information from the refference to avoid misunderstanding

Line 504-510

Telopeptides that contain large amounts of hydrophobic residues greatly assist the inter-microfibrillar interaction. In addition, ASC's inter- and intra-molecular cross-linking are richer than those of PSC [139]. The presence of telopeptide also affects thermal stability; for instance, thermal stability between PSC and ASC ranges from 1–4 °C [110,135,140–143], but in the skin of grass carp collagen, the difference in denaturation temperature of PSC (telopeptide removed) and ASC (telopeptide retains) is 6.84 °C with superiority held by collagen that retains telopeptide [144].

Comment:

Line 515. “Cross-linking is a method of fabricating collagen biomaterials that connects collagen fibrils through chemical bonds into fibers” According to our research, the cross-linking of collagen might result in the fibrious network.

Response:

Thank you for your remarks, the sentence now accounts the continuation of fiber into a fibrous network

Line 566-569

Cross-linking is a method of fabricating collagen biomaterials that connects collagen fibrils through chemical bonds into fibers as precursor to a fibrious network, which are subsequently suitable as biomaterials such as sponges, hydrogels, or membranes.

Reviewer 3 Report

The work is described in detail. The work is modern and interesting for publication in this journal. I have no special recommendations, as the authors have considered all the important issues.

Although it would be interesting to know the opinion of the authors on the question - why are there still so few biomaterials from marine collagens approved for real clinical practice?

Author Response

Responses to Reviewer 3

Manuscript ID Number: marinedrugs-2089531

Manuscript Title: Cutting Edge Aquatic-based Collagens in Tissue Engineering

We thank the referees for their time spent carefully reviewing the manuscript. Below are our responses to the comments of reviewers. We have copied and pasted their comments. The reviewer’s as well as editor comments appear in italic and follow in the same order as written by the referee. Our responses appear in regular font.

Comment:

The work is described in detail. The work is modern and interesting for publication in this journal. I have no special recommendations, as the authors have considered all the important issues.

Although it would be interesting to know the opinion of the authors on the question-why are there still so few biomaterials from marine collagens approved for real clinical practises?

Response:

We have updated the paper with our opinion about why the use of marine collagen as a biomaterial for real clinical is still few. In summary, there are two main reasons which are (1) large-scale production of AQCols still has several barriers; for example, the cold chain in the production of marine collagen and the preservation of marine waste is a critical point that makes the cost of production more expensive; furthermore, maintaining the supply of raw materials is also critical because the large-scale use of non-cultivated marine organisms could be limited by the sustainability feedstock (Line 869-890).

The second reason is (2) The potentiality of marine collagen has still not been explored yet, so we need more researchers to research marine collagen scaffolds' fabrication with various methods, clinical studies to evaluate aspects related to safety and efficacy, research about collagen-based bioink for different tissue-engineering applications and regenerative medicines (especially the combination of collagen-based bioink and 3D bioprinting) (Line 947-965).

Round 2

Reviewer 2 Report

I thank the authors for revising the manuscript. All the questions had been answered and I have no more questions. 

  •